# Metastability and Ostwald step rule in the crystallisation of diamond and graphite from molten carbon

**Davide Donadio** [1] ✉, **Margaret L. Berrens** [1,2], **Wanyu Zhao** [3], **Shunda Chen** [3] & **Tianshu Li** [3]

Experimental challenges in determining the phase diagram of carbon at temperatures and pressures near the graphite-diamond-liquid triple point are often related to the persistence of metastable crystalline or glassy phases, superheated crystals, or supercooled liquids. A deeper understanding of the crystallisation kinetics of diamond and graphite is crucial for effectively interpreting the outcomes of these experiments. Here, we reveal the microscopic mechanisms of diamond and graphite nucleation from liquid carbon through molecular simulations with first-principles machine learning potentials. Our simulations accurately reproduce the experimental phase diagram of carbon near the triple point and show that liquid carbon crystallises spontaneously upon cooling. Metastable graphite crystallises in the domain of diamond thermodynamic stability at pressures above the triple point. Furthermore, whereas diamond crystallises through a classical nucleation pathway, graphite follows a two-step process in which low-density fluctuations forego ordering. Calculations of the nucleation rates of the two competing phases confirm this result and reveal a manifestation of Ostwald's step rule, where the strong metastability of graphite hinders the transformation to the stable diamond phase. Our results provide a key to interpreting melting and recrystallisation experiments and shed light on nucleation kinetics in polymorphic materials with deep metastable states.

The crystallisation of carbon from the melt under extreme conditions is highly relevant to Earth and planetary science[1,2], materials manufacturing, and nuclear fusion research[3]. The thermodynamic conditions near the graphite-diamond-liquid (GDL) triple point are especially of interest for geological and technological applications, but high-pressure flash-heating experiments aiming to resolve this region of the phase diagram of carbon exhibit large discrepancies[4–9]. The diverse technological applications of carbon materials stem from the different types of covalent bonding with tetrahedral patterns in diamond and tetrahedral amorphous carbon on one hand,

and hexagonal motifs in graphite or graphitic carbon on the other. Such differences in covalent bonding engender large metastability of both common crystalline phases, graphite and diamond. These chemical features dictate the phase diagram of carbon, which exhibits a GDL triple point at ~12 GPa and 4500 K. Whereas the diamond-in-the-sky hypothesis[2] and recent experiments on nuclear fusion by laser shock compression[3] have prompted several studies of the phase diagram of carbon and hydrocarbons at extreme pressures, the GDL triple point region has not been thoroughly addressed either by experiments or simulations. Conversely, understanding the

[1]Department of Chemistry, University of California Davis, Davis, CA, USA. [2]Quantum Simulations Group, Physics Division, Lawrence Livermore National Laboratory, Livermore, CA, USA. [3]Department of Civil and Environmental Engineering, George Washington University, Washington, DC, USA. ✉e-mail: ddonadio@ucdavis.edu

thermodynamics and kinetics of phase changes in this region is of paramount importance for materials and planetary sciences. For example, in geosciences, diamonds reveal tectonic processes in Earth-like planets and the deep-earth cycle of carbon and other light elements[1]. In materials processing, the liquid phase is an intermediate in the laser synthesis of carbon materials, including artificial diamonds, nanodiamonds, and N-vacancy doped diamonds for quantum computing, and its crystallisation behaviour upon cooling determines the properties of the synthesis products[10]

Ostwald hypothesised that nucleation from the melt does not necessarily proceed directly to the thermodynamically most stable phase but through the phase that is separated by the lowest free energy barrier from the liquid[11]. This effect significantly accelerates crystallisation rates in the presence of metastable critical points[12]. Molecular simulations of nucleation from the melt confirmed Ostwald's step rule hypothesis in Lennard-Jones liquids, colloidal particles, and protein crystallisation[13–15]. In the latter case, a two-step process was observed experimentally using dynamic light scattering and time-resolved spectroscopy[16]. Non-classical multi-stage nucleation pathways have also been discussed in the context of mineralisation from solution, with abundant numerical and experimental evidence[17], and in the case of polymorphic materials, the precipitation of metastable phases has been observed[18]. Nevertheless, it is unexpected to encounter such a rich phenomenology in the homogeneous nucleation of a simple monoatomic system like carbon.

In this work, we investigate the kinetic aspects of diamond and graphite crystallisation, probing homogeneous nucleation from molten carbon at moderate pressures using machine-learning-accelerated, quantum-accurate molecular dynamics simulations. The range of pressures considered, between 5 and 30 GPa, corresponds to the pressure of the Earth's mantle between 120 and 750 Km. Our simulations show the unexpected spontaneous crystallisation of metastable graphite at pressures up to 15 GPa, well within the domain of thermodynamic stability of diamond, and unravel the fundamentally distinct molecular mechanisms that lead to the nucleation of diamond and graphite. The crystallisation pathways exhibit a combination of these effects that are the consequence of Ostwald's step rule and the large free energy barrier between graphite and diamond. These observations open new perspectives in the theory of crystallisation of single-component systems with metastable phases.

## Results

### Phase diagram and spontaneous crystallisation

We first discuss the phase diagram of carbon in the region of the GDL triple point (Fig. 1a). Experimentally, the triple point was initially determined at $T = 4100 \pm 200$ K, and $P = 12.5 \pm 1$ GPa by extrapolating the graphite-diamond coexistence line to high temperatures[19]. Extrapolating the melting curves of graphite[5] and diamond[4] suggests that the triple point should be at a higher temperature, between 4400 and 4800 K. However, in general, the experimental determination of the GDL triple point entails large uncertainties[20]. In contrast to earlier measurements[21], experiments now agree that the melting line of diamond has a positive slope while that of graphite has a negative slope at the triple point. The melting temperature of graphite also exhibits a maximum at ~5 GPa. This was initially thought to be indicative of a liquid-liquid phase transition[22], which was not confirmed by further simulations or experiments[23,24]. Recent experiments suggest that the melting point of graphite at 2 GPa and that of diamond at 15 GPa may be over 6000 K, significantly higher than in previous works, adding further uncertainty to the carbon phase diagram and the location of the GDL triple point[7,8].

Here we compute the graphite-liquid and diamond-liquid coexistence lines, along with their metastable extensions, by large-scale phase-coexistence MD simulations using feed-forward neural network potentials fitted with an evolutionary strategy (neuroevolution

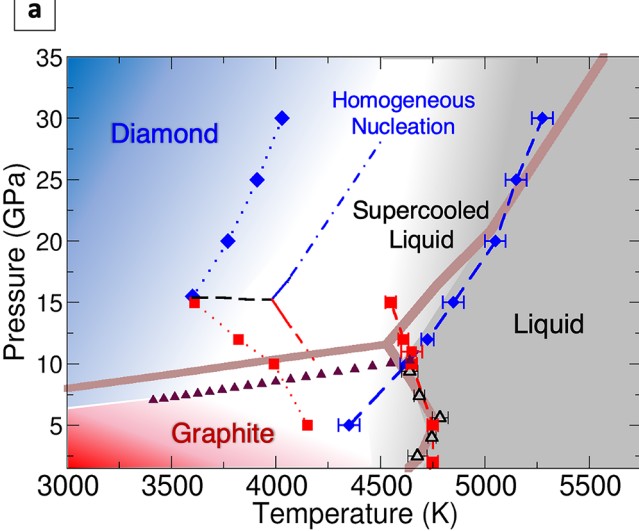

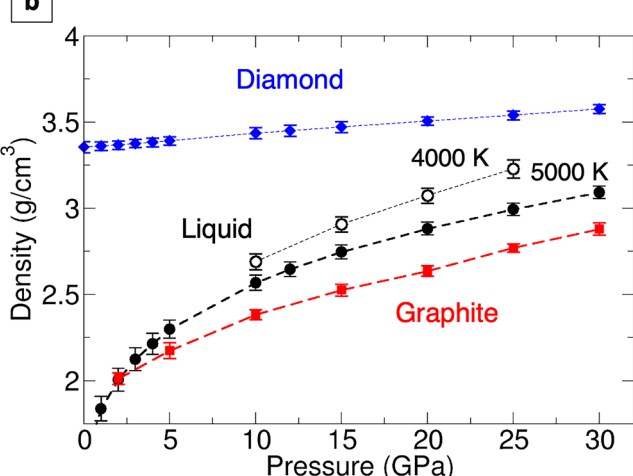

**Fig. 1 | Phase diagram and density of carbon phases. a** Phase diagram of carbon in the moderate pressure range 2–35 GPa. The melting temperatures of diamond (blue diamonds) and graphite (red squares) computed by phase-coexistence molecular dynamics are represented by symbols with error bars. Error bars are computed as the difference between the lowest temperature at which the crystalline phase melts and the highest temperature at which it grows. The graphite/diamond phase boundary (maroon triangles) is obtained by Gibbs-Duhem integration starting from the triple point[30]. The experimental phase diagram (brown line) combines results from refs. 4,5. (black triangles). The temperatures at which we observed spontaneous crystallisation of diamond (blue) and graphite (red) are indicated with the respective symbols without error bars. The dashed black line indicates the maximum pressure at which graphite nucleates faster than diamond. The dashed-dotted lines indicate the homogeneous nucleation temperature for graphite (red) and diamond (blue), defined as the temperature at which crystal nucleation is faster than $10^{14}$ m$^{-3}$ s$^{-1}$. **b** Density of diamond, graphite at 5000 K, and liquid carbon at both 4000 and 5000 K at different pressures. Error bars indicate the standard deviation of the density during the molecular dynamics runs.

potential, NEP)[25] and trained on energies and forces computed at the level of density functional theory (DFT) either in the local density approximation (LDA)[26] or with a generalised gradient approximation (GGA) functional with non-local van der Waals correlation 'OptB88-vdW'[27,28]. We dub these two models NEP@LDA and NEP@OptB88-vdW.

Our simulations closely reproduce the most commonly accepted experimental phase diagram regardless of the chosen NEP model[4,29]. In the NEP@LDA diagram, the intersection between the two melting

curves establishes a GDL triple point at $P = 10.5 \pm 0.5$ GPa and $T = 4650 \pm 50$ K, at a slightly lower pressure than in most experiments but well within the temperature range. Starting from the triple point, we computed the graphite-diamond coexistence line using the Gibbs-Duhem integration method[30]. The diamond-liquid coexistence line agrees with former FPMD simulations[24,31]. We note that our model reproduces the reentrant melting line of graphite in excellent agreement with measurements[5]. The reason for the negative slope of the melting line of graphite near the triple point is that the density of liquid carbon is higher than that of graphite at pressures higher than 2 GPa (Fig. 1b). The density crossover at 2 GPa corresponds to the change of slope of the graphite melting line from positive to negative. This characteristic is extremely consequential for the crystallisation behaviour of graphite and diamond and is rarely captured by either empirical potentials or machine learning models[32], with few notable exceptions, such as the revised long-range carbon bond-order potential (LCBOPII) and the environment-dependent interatomic potential[33–35]. Notably, using different parameterisations of NEP does not have a significant effect on the melting lines and the location of the triple point. The phase diagrams reported in Fig. S1 share the same features as those in Fig. 1.

Since the approximations in the LDA exchange and correlation functional entail inaccuracies in the equations of state and the phase diagram of materials, usually overestimating the density of both solids and liquids[36], we repeated the calculation of the phase diagram of carbon with NEP@OptB88-vdW[27,28,37]. The NEP@LDA and NEP@OptB88-vdW phase diagrams are reported side by side in Fig. S2. The GDL triple point with the NEP@OptB88-vdW model is at the same temperature as NEP@LDA but at a higher pressure ($P = 17$ GPa and $T = 4660$ K). In general, the two-phase diagrams exhibit the same features with just a pressure offset of  ~6.5 GPa.

As expected from previous DFT calculations[36], the equations of state of liquid carbon, computed at 5000 and 7000 K differ, with NEP@LDA overestimating and NEP@OptB88-vdW underestimating the density compared to experiments (Fig. S3)[7]. However, the structure of the liquid, characterised through radial distribution functions (Fig. S4) and the analysis of the local coordination (Fig. S5), exhibit the same behaviour as the pressure is increased. At any pressure between 5 and 30 GPa, compared to the NEP-OptB88-vdW liquid, the NEP@LDA liquid exhibits a larger number of tetrahedrally coordinated carbons that increases linearly with the pressure, and a lower number of two-fold coordinated carbons that decreases. The number of three-fold coordinated carbons has a maximum between 5 and 10 GPa for NEP@LDA and above 15 GPa for NEP@OptB88-vdW. The differences in the properties of the liquids obtained with the two models may be reconciled with a pressure shift of about 10 GPa. The liquid structure is consistent with previous studies based on DFT or accurate empirical potentials[32,33,38].

Due to the small size of molecular models and the use of periodic boundary conditions, MD simulations customarily overestimate the stability of liquid phases below the melting point and the formation of amorphous phases by quenching from the melt. However, here, liquid carbon models that are first equilibrated at 5000 K and then quenched at the rate of 60 K/ns exhibit spontaneous crystallisation into either diamond or graphite, depending on the pressure. The temperatures at which spontaneous crystallisation was observed are reported in the phase diagrams (Figs. 1a and S2) and define the lowest-temperature boundary of the region of metastability of supercooled liquid carbon. This leads us to conclude that liquid carbon, as opposed to other group IV elemental liquids like silicon and germanium, is not a good glass former at high pressure. Therefore, quenching from the melt at high pressure is not a viable route to obtain tetrahedral amorphous carbon. The temperature of spontaneous crystallisation of graphite decreases with pressure with a more accentuated negative slope than the phase-coexistence line. Conversely, the temperature of

spontaneous crystallisation of diamond increases with pressure but with a similar slope as the melting curve.

Strikingly, we observe spontaneous crystallisation of graphite at pressures as high as 15 GPa, which is about 5 GPa higher than the pressure of the GDL triple point and 7 GPa higher than the graphite-diamond phase boundary at the crystallisation temperature. The occurrence of a metastable structure in the crystallisation pathway is predicted by Ostwald's step rule when the metastable phase has a lower free energy barrier than the thermodynamically stable one[11,39]. This condition corresponds to a closer structural similarity between the nucleating metastable structure and the parent liquid. We observe that the highest pressure at which graphite crystallises spontaneously corresponds to the condition where the density of the liquid is midway between that of graphite and that of diamond (see Fig. 1b). This observation suggests that the density of the liquid is an important driver of crystallisation. In turn, the change of preference for spontaneous crystallisation from graphite to diamond is not associated with any abrupt changes in the structure of liquid carbon: the equation of state, pair correlation function, and local coordination undergo gradual changes as the pressure increases (Figs. S3–S5) in accordance with previous studies[33,38]. It is worth noting that at these pressures, the liquid is still predominantly three-fold coordinated. In most cases of manifestation of Ostwald's step rule, the metastable nucleus transforms spontaneously into the stable phase as it grows. In our simulations, however, graphite persists, as the activation free energy to transform graphite into diamond is high, and crystallisation happens too rapidly. The crystallisation of metastable crystalline phases from rapidly quenched liquids has been observed for minerals and alloys, especially when simple low-density structures compete with complex structures, but it was not formerly hypothesised for carbon. The pressure differences between the graphite-diamond phase coexistence line and 15 GPa are insufficient to overcome the high barrier to transforming graphite into diamond over the typical MD timescales[40,41]. However, experimentally graphite transforms directly into diamond upon compression at 12.5 GPa above 3000 K[42], thus suggesting that graphite would be a long-lived metastable intermediate state but should eventually transform into diamond.

Spontaneous crystallisation of metastable graphite above the graphite-diamond coexistence line also occurs in simulations performed with the NEP@OptB88-vdW model (Fig. S2). In this case, we observed metastable nucleation of graphite up to 19 GPa at 3670 K, where the corresponding graphite-diamond phase boundary is at 14.5 GPa (Fig. S2). Since the two NEP models provide the same qualitative physical picture of the thermodynamics of carbon in the region of interest and of the kinetics of crystallisation, we restrict the analysis in the following sections to results obtained using the NEP@LDA model.

## Nucleation rates

To validate the observed metastable crystallisation of graphite from liquid carbon, we computed the nucleation rates of graphite and diamond at moderate supercooling. These calculations require the use of an enhanced sampling technique with the possibility of biasing the nucleation of either phase. For this purpose, we used forward flux sampling (FFS) molecular dynamics[43,44]. To identify the graphite and diamond nucleating crystallites, we adopted local order parameters based on the spherical harmonic expansion of the atomic environment, $q_l$[45,46]. For diamond, we use $l = 6$ and for graphite $l = 3$ with the additional constraint that atoms must be three-fold coordinated. Figure 2d−f shows the nucleation rates of graphite and diamond as a function of temperature up to 4100 K at $P = 13$, 15, and 17 GPa. These calculations confirm a preference for graphite crystallisation at any temperature at 13 GPa. Nucleation rates for the two phases are similar at 15 GPa with a slight preference for graphite, whereas crystallisation of diamond is predominant at 17 GPa.

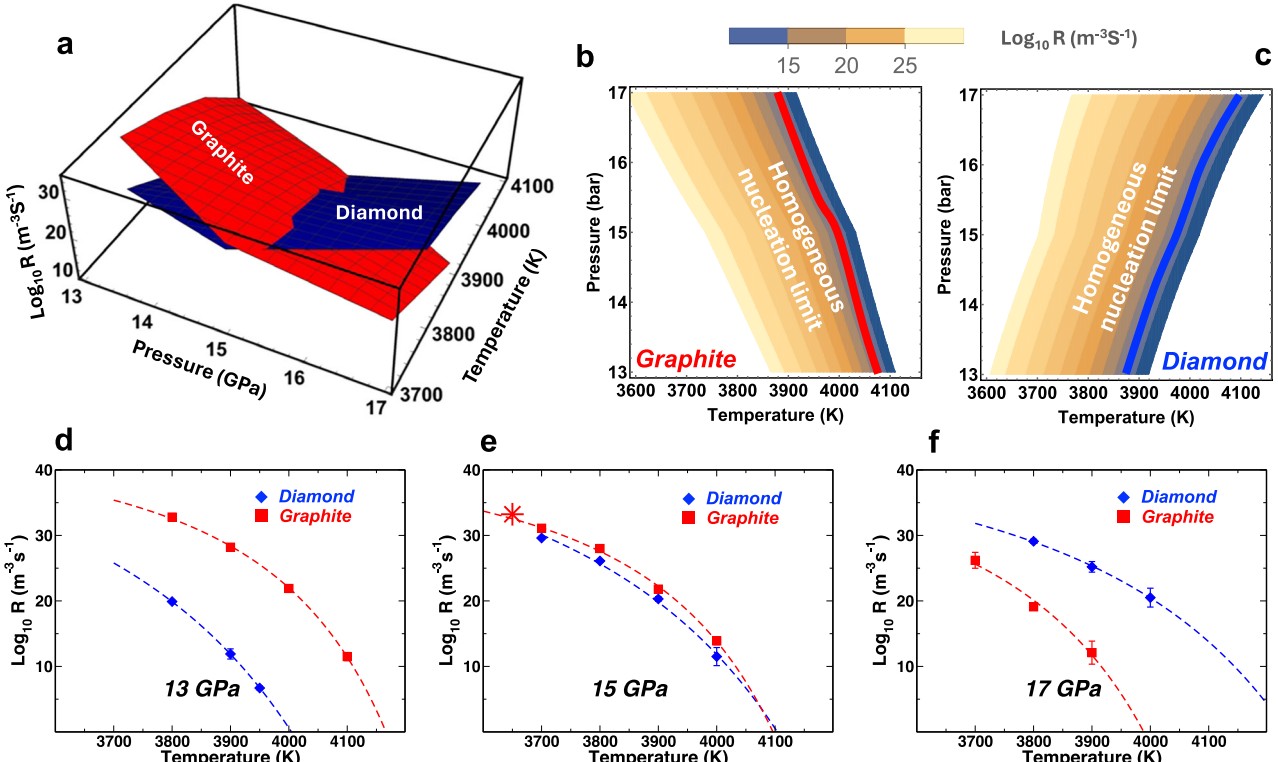

**Fig. 2 | Nucleation rates of graphite and diamond. a** A 3D view of log base 10 rate (m$^{-3}$ s$^{-1}$) as a function of both temperature and pressure shows graphite and diamond yield similar nucleation rates at ~ 15 GPa over a range of temperature. In the contour of the log base 10 rates for **b** graphite and **c** diamond, the iso-rate lines are nearly parallel. Solid lines on the contour indicate the homogeneous nucleation limit (defined as nucleation with a rate of $10^{14}$ m$^{-3}$ s$^{-1}$). **d–f** The calculated nucleation rates through forward flux sampling (solid symbols) for both graphite and diamond can be fitted well against classical nucleation theory (dashed lines) at 13, 15, and 17 GPa, respectively. Each data point represents the geometric mean nucleation rate obtained from three independent FFS runs, with error bars being the standard errors of the mean. The star in **e** stands for the nucleation rate of graphite obtained by direct simulation at 3650 K and 15 GPa.

The rates and the estimates of the critical size of the nuclei obtained by FFS allow us to fit a model based on classical nucleation theory (CNT). In CNT, the free energy of formation of a crystalline nucleus of size $N$ in a liquid is the result of a negative term that accounts for the difference in chemical potential between the solid and the liquid ($\Delta\mu$) and a positive term that accounts for the cost of creating solid/liquid interface. The CNT rate can be expressed as:

$$R_{CNT}(T) = A\exp\left(-\frac{\Delta G^{\star}}{k_B T}\right) \qquad (1)$$

where $A$ is a kinetic prefactor and $\Delta G^{\star}$ is the free energy of formation of the critical nucleus:

$$\Delta G^{\star} = \frac{16\pi\gamma_{LS}^{3}}{3\rho^2\Delta\mu^2} \qquad (2)$$

In this expression, $\rho$ is the number density of the solid, and $\gamma_{LS}$ is the average interfacial tension between the liquid and the solid phase. $\Delta\mu$ is usually approximated as $\Delta\mu = \Delta H_m(T_m - T)/T$ where the enthalpy of melting ($\Delta H_m$) and the melting temperature ($T_m$) are calculated from MD simulations at solid/liquid coexistence conditions, leaving $A$ and $\gamma_{LS}$ as the only unknowns. Figure 2d–f shows that CNT (continuous lines) fits the FFS nucleation rates of both graphite and diamond. We note that the fitting of the FFS rate against CNT does not rely on the assumption of a spherical critical nucleus, but only requires the shape to remain unchanged at different temperatures. $\Delta G^{\star}$ obtained from this fit provides an estimate of the effective liquid/solid interfacial tension for graphite and diamond nuclei assumed spherical, where the former is $\gamma_{LS,G} = 1.15$ J/m$^2$ and the latter is $\gamma_{LS,D} = 2.0$ J/m$^2$ at $P = 15$ GPa. At

an atomistic level, the high value of $\gamma_{LS,D}$ at moderate pressure can be explained by the stark difference in the bonding environment between the liquid, mostly three-fold coordinated, and the four-fold coordinated diamond crystallite[46]. The much smaller value of $\gamma_{LS,G}$ suggests that the direct nucleation of diamond can occur only when the chemical potential term is dominant, i.e., when the liquid is substantially more supercooled with respect to diamond than to graphite.

Fitting the CNT model on rates at different pressures provides an estimate of the thermodynamic conditions for metastable crystallisation of graphite (Fig. 2a). In analogy with other cases in which liquids can be supercooled well below the melting line, e.g., water, we can define a homogeneous nucleation limit for supercooled liquid carbon as the temperature at which $R < 10^{14}$ m$^{-3}$s$^{-1}$. The homogeneous nucleation lines and the line at which graphite and diamond nucleation rates are equal (black dashed line) are also represented in Fig. 1a.

## Nucleation pathways
To study unbiased nucleation pathways and obtain an estimate of the nucleation rates of graphite and diamond at deep supercooling[47], we have performed direct MD simulations at pressures of 15 GPa and 15.5 GPa and a temperature of 3650 K. Averaging over 10 runs at each pressure, we computed a mean first passage time (MFPT) $\tau_{MFPT} = 21.1 \pm 2.8$ ns corresponding to a nucleation rate of $1.7 \pm 0.2 \times 10^{33}$ m$^{-3}$s$^{-1}$ for graphite at 15 GPa, and $\tau_{MFPT} = 16.3 \pm 5.4$ ns corresponding to $2.3 \pm 0.7 \times 10^{33}$ m$^{-3}$s$^{-1}$ for diamond at 15.5 GPa. The (MFPT) nucleation rate of graphite is found to match well the CNT model fitted through the FFS rates (Fig. 2e), further confirming the validity of CNT and the applicability of FFS for graphite nucleation.

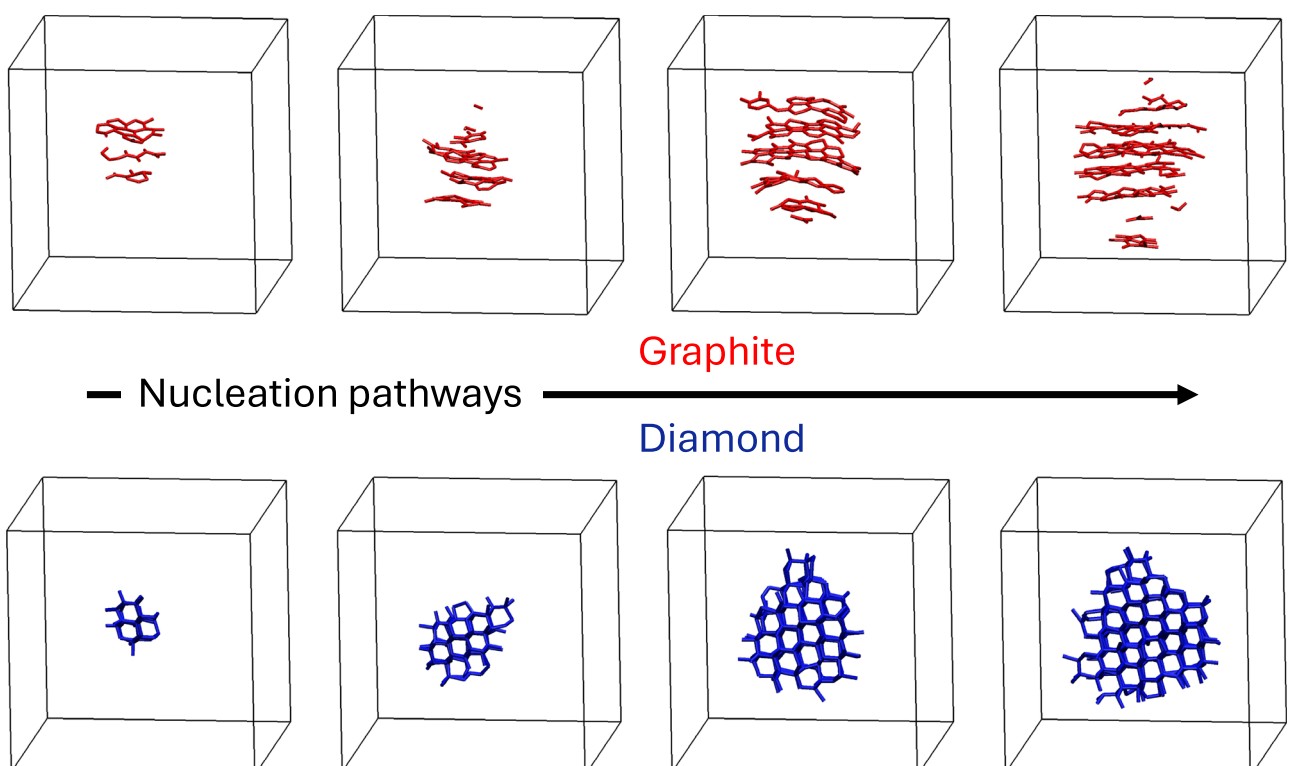

**Fig. 3 | Crystal nuclei.** Nucleation pathways of graphite (top row) and diamond (bottom row) from direct molecular dynamics simulations at pressures of 15 and 15.5 GPa and a temperature of 3650 K. Bonds are coloured according to the value of the local order parameter.

Comparing the qualitative features of the growing nuclei in direct simulations to FFS confirms that the chosen order parameters to define the flux interfaces do not bias the sampling towards unrealistic nucleation pathways. Figure 3 shows snapshots along the crystallisation pathway of graphite and diamond from unbiased simulations in which liquid carbon is cooled at 15 and 15.5 GPa. Diamond nucleates into compact crystallites with an isotropic aspect ratio, which justifies the spherical nucleus approximation often adopted in CNT. Similar behaviour was observed in the crystallisation of other tetrahedral systems, including silicon, germanium, and water ice[44,48]. This also means that the average estimate of $\gamma_{LS,D}$ is reliable. Conversely, graphite nucleation proceeds through the formation and stacking of small graphite patches which produce anisotropic crystallites elongated along the cross-plane axis of graphite (Fig. 3top). The growing graphite crystals are highly non-spherical, and the free energy of the interface between the different facets and the liquid dictates their shape.

Employing the same order parameters used to analyse the crystallisation trajectories, we calculated the roughness of the interface between liquid carbon and the basal (0001) and primary prismatic (11$\bar{2}$0) planes of graphite in the two-phase simulations used to compute the melting line. From this roughness, it is possible to estimate the interfacial free energy $\gamma_{LS,G}$ for each surface, yielding $1.3 \pm 0.08$ Jm$^{-2}$ for the basal plane and $0.73 \pm 0.07$ Jm$^{-2}$ for the prismatic plane. The lower interfacial tension estimate of the primary prismatic facet is consistent with the findings of the direct MD and FFS simulations, where graphite was found to nucleate through the formation of anisotropic crystallites elongated along the cross-plane axis of graphite that resemble the primary prismatic facet of graphite. The basal plane $\gamma_{LS,G}$ is in agreement with previous estimates from a continuum theory of carbon phases fitted to thermodynamic data (1.6 Jm$^{-2}$)[49,50]. However, these theories cannot grasp the large differences among planes which dictate the nucleation pathways at the molecular scale.

**Prenucleation mechanism**

Previous considerations indicate that density is the main similarity parameter that drives the crystallisation of graphite or diamond from molten carbon, since both transitions involve large density variations. At the tipping pressure of 15 GPa, the crystallisation of graphite is accompanied by a density drop from 2.9 to 2.5 g/cm³, whereas when diamond is formed, the density rises to 3.5 g/cm³. Such extreme density differences between the liquid and the solid phases affect nucleation pathways. Previous studies showed that, following Ostwald's step rule, nucleation could be a two-step process in which density fluctuations with the formation of high-density amorphous or liquid aggregates precede crystalline ordering. This effect was observed during crystal nucleation both in supercooled single-component liquids and supersaturated solutions[14,15,51–53]. We observe a similar effect in the crystallisation of graphite. Figure 4a shows the probability that low-density clusters and crystalline nuclei of a given size form spontaneously in the liquid at 3650 K and 15 GPa. The shape of the distribution indicates that the crystallinity and low-density order parameters do not correlate. The occurrence of crystalline clusters without local low-density fluctuations can be explained by the fact that the system tends to form diamond-like small nuclei. However, spontaneous nucleation pathways to graphite, one of which is represented in the graph as a red dotted line, evolve through the formation of a low-density cluster that eventually develops into a crystalline nucleus. In the case of diamond nucleation, instead, as shown in Fig. 4b, the density fluctuations producing high-density clusters are correlated to the crystallinity order parameter. Consequently, the crystalline nucleus is not preceded by a higher-density disorder precursor: local densification and ordering coincide. This difference further supports the evidence that graphite nucleation follows a lower-barrier free energy pathway, facilitated by local low-density fluctuations in liquid carbon even at the thermodynamic conditions at which diamond is the stable phase.

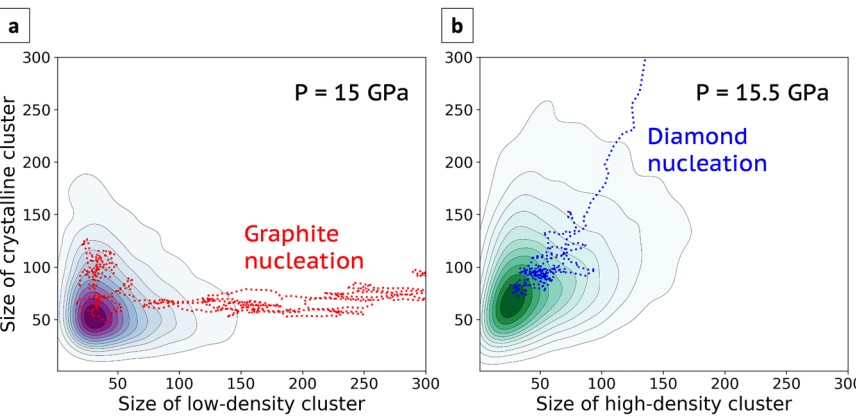

**Fig. 4 | Nucleation mechanism in the order parameters space.** The two contour plots show the logarithmic probability distribution of the size of clusters of carbon atoms selected according to their density and crystallinity order parameter in liquid carbon at 3650 K. **a** The $x$ axis indicates the size of the biggest low-density cluster ($\rho < 2.45$ g/cm$^3$ within a sphere of 3 Å radius) at a pressure of 15 GPa, and in **b** of the biggest high-density cluster ($\rho > 3.4$ g/cm$^3$ within a sphere of 3 Å radius) at a pressure of 15.5 GPa. The dotted lines represent spontaneous nucleation pathways of graphite (**a**) and diamond (**b**).

## Discussion

Accurate and efficient machine learning potentials trained on density functional theory calculations, enabling a total of several microseconds of MD simulations of systems of several thousand atoms, provided us with an invaluable tool to explore the phase diagram of carbon near the GDL triple point and the intricate kinetics of homogeneous nucleation of graphite and diamond from the melt. Whereas liquid carbon is often considered a glass former, we observed spontaneous crystallisation of both graphite and diamond in unbiased MD simulations in which the liquid is cooled rapidly at constant pressure. This suggested that liquid carbon at pressures between 5 and 30 GPa is not a glass former, and provided us with unbiased trajectories to examine the nucleation pathways at the molecular scale. We observed an unexpected wealth of physical phenomena related to homogeneous nucleation in a deceivingly simple monoatomic system such as liquid carbon. These phenomena are inherently connected to Ostwald's step rule, leading to highly non-classical nucleation effects. In MD simulations, graphite crystallises spontaneously in the domain of thermodynamic stability of diamond due to the smaller density differences between liquid carbon and graphite and the lower liquid/solid interfacial free energy. Depending on the details of the potential, metastable graphite may crystallise up to a pressure between 5 and 7.5 GPa above the graphite-diamond coexistence line. In this range of pressures, graphite is a long-lived metastable state on the way to the crystallisation of diamond[42]. Additionally, we found graphite nucleates through a non-classical two-step process where crystalline ordering is preceded by the formation of a low-density liquid region that further lowers the barrier to nucleation. Large differences in the interfacial free energy of different facets dictate the strongly anisotropic shape of growing graphite crystallites. Conversely, at higher pressure, diamond nucleates through a classical one-step process, forming densely packed tetrahedrally bonded crystalline spherical clusters. The two intrinsically different nucleation mechanisms of graphite and diamond are schematically represented in Fig. 5. Even if graphite nucleation is non-classical, a CNT fit of the nucleation rates yields physically meaningful parameters for both graphite and diamond, suggesting that even non-classical nucleation processes can be mapped on a CNT model[54].

The insight into homogeneous nucleation from carbon melt provided by our simulations may help resolve inconsistencies among historical electrical and laser flash-heating experiments aimed at resolving the phase diagram of carbon near the GDL triple point. Depending on the details of these experiments and the recrystallisation conditions, the system may remain trapped in metastable graphitic configurations. These observations may also impact the manufacturing of carbon-based materials such as synthetic diamonds and nanodiamonds at high pressure and high temperature.

## Methods

### Machine-learning potentials

Molecular dynamics simulations were performed using the GPUMD v3.6 code with a NEP3 neuroevolution potential[25,55] fitted to DFT calculations in the LDA for the exchange and correlation functional (NEP@LDA) and a NEP4 potential developed based on the OptB88-vdW functional (NEP@OptB88-vdW)[27,28]. For both potentials, the databases of configurations, energies, forces, and virials were formerly generated to study diverse carbon systems, including graphite, diamond, low and high-density carbon, and liquid carbon[26,28].

We conducted extensive tests to verify that the resulting NEP potentials are suitable for modelling diamond and graphite. Besides the phase diagrams shown in Figs. 1 and S2, we verified that the NEP@LDA reproduces the structural, vibrational, and elastic properties of diamond and graphite accurately (Table S1). These results suggest that, whereas LDA is the lowest-level approximation for exchange and correlation in DFT, it accurately describes the mechanical and thermodynamic properties of graphite, particularly the liquid-graphite phase coexistence. In the pressure range of interest, i.e., above 10 GPa, our predictions for the melting line of diamond are within 100 K of DFT calculations with GGA functionals[24,31].

Additionally, we verified the sensitivity of our predicted phase diagram and spontaneous crystallisation conditions upon the hyperparameters and the training process of the NEP@LDA potentials. We generated three more carbon NEP@LDA potentials varying several hyperparameters, including |cutoff| (cutoff radii for the radial and angular descriptor components), |n_max| (number of radial functions for the radial and angular descriptor components), |basis_size| (number of radial basis functions for the radial and angular descriptor components), population size, and number of generation. Figure S1 shows that the melting line of graphite may shift by at most 100 K whereas the melting line of diamond is insensitive to the different NEP parameterisations. Accordingly, crystallisation temperatures undergo variations up to 150 K including the uncertainty inherent to the stochastic nature of this process. The highest pressure at which metastable graphite crystallises spontaneously varies between 13 and 15 GPa. The hyperparameters for training the NEP@LDA and NEP@OptB88 models are reported in Table S2.

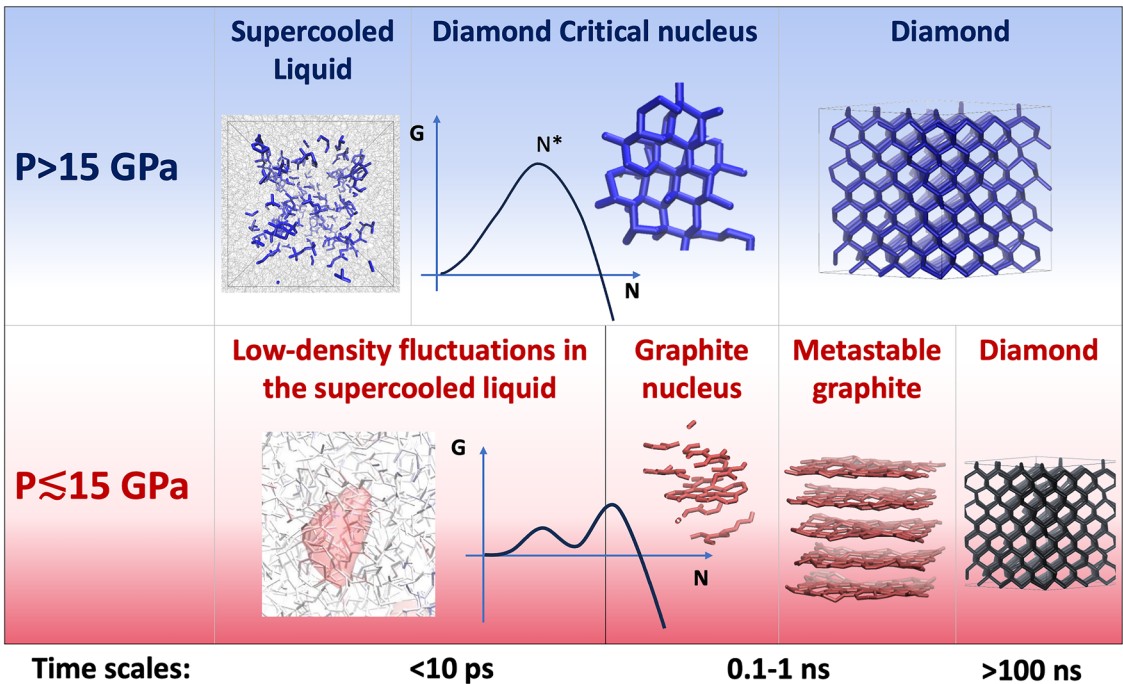

**Fig. 5 | Classical and non-classical crystallisation pathways.** Schematic representations of the one-step classical nucleation pathway of diamond at high pressure and the non-classical pathway that leads to the metastable crystallisation of graphite at $P \lesssim 15$ GPa, which will eventually transform into diamond through a solid-solid phase transition.

## Molecular dynamics

MD simulations probing the properties of liquid carbon and spontaneous crystallisation were performed using the GPUMD package for cubic systems containing 4096 carbon atoms[55]. The equations of motion were integrated with a timestep of 0.5 fs. We have verified that with a 0.5 fs timestep, MD simulations of liquid carbon at ~4500 K conserve the energy within 1;meV/atom over 100 ps (Fig. S6). The isobaric canonical ensemble (constant number of particles, pressure, and temperature - NPT) was controlled using the stochastic rescaling scheme with coupling times of 1 ps for the temperature and 5 ps for the pressure[56,57]. Spontaneous crystallisation of diamond and graphite was observed in direct MD simulations of liquid carbon at constant pressure, in which the temperature was ramped from 5000 K to 3500 K in 25 ns, corresponding to a cooling rate of 60 K/ns.

These simulations were initiated from liquid models equilibrated at 5000 K for at least 5 ns. The structural features of these liquids are characterised by the radial distribution functions and the coordination numbers reported in Figures S4 and S5.

## Calculation of the phase boundaries

We used phase coexistence simulations to determine the phase coexistence temperatures of liquid carbon with diamond and graphite[31,58]. These runs were set up by joining a diamond/graphite simulation cell with a liquid carbon simulation cell equilibrated at 5000 K. Liquid carbon was put in contact with the (100) facet of diamond and the (11$\bar{2}$0) facet of graphite (prismatic plane with armchair configuration). Two-phase simulations of graphite crystal growth and melting constructed with the liquid in contact with the basal plane (0001) exhibited slower growth/melting dynamics, proving less useful to determine the melting point. These simulations comprise systems of about 10,000 carbon atoms, thus avoiding issues with size effects. Snapshots of liquid/graphite and liquid/diamond two-phase configurations are displayed in Fig. S7. For each pressure, we ran a series of anisotropic NPT simulations with independent edges and fixed angles at

temperatures chosen every $\Delta T = 50 K$. We define the melting point as the temperature between two simulations at $T$ and $T + \Delta T$, where the crystalline phase grows ($T$) and shrinks ($T + T$).

To compute the graphite-diamond phase boundary and to refine the solid-liquid coexistence lines, we integrated the Gibbs-Duhem equation[30] by finite differences in $\beta$:

$$\frac{dp}{d\beta} = -\frac{\Delta h}{\beta \Delta v} \tag{3}$$

where $\beta = 1/k_B T$, $\Delta h$ is the difference in molar enthalpy between the two phases, and $\Delta v$ is the difference in molar volume. For the graphite-liquid coexistence line, we integrated Eq. (3) starting from the triple point with a fixed $\Delta\beta = 0.05$ eV$^{-1}$. At each step, $\Delta h$ and $\Delta v$ are estimated as ensemble averages over the second half of 100 ps NPT runs.

## Nucleation rates from forward flux sampling and order parameters

Nucleation rates of diamond and graphite are computed using our implementation of forward flux sampling (FFS) in LAMMPS, based on carbon NEP. In FFS, nucleation rate $R$ is given as the product of $\Phi_{\lambda_0}$, which is the initial flux rate of nucleation trajectory crossing the initial milestone $\lambda_0$, and $P(\lambda_F|\lambda_0)$, which measures the probability for a trajectory starting from $\lambda_0$ and successfully reaching the final milestone $\lambda_F$. $\Phi_{\lambda_0}$ is obtained by $N_0/t_0 V$, where $N_0$ (120) is the number of successful crossings collected at $\lambda_0$, $V$ is the volume of the simulation cell, $t_0$ is the total time for $N_0$ crossings. $P(\lambda_F|\lambda_0) = \prod_{i=1}^{n} P(\lambda_i|\lambda_{i-1})$, where $P(\lambda_i|\lambda_{i-1}) = N_i/M_{i-1}$ is computed through conducting $M_{i-1}$ MD trial runs starting from the milestone $\lambda_{i-1}$ and collecting $N_i$ (120) successful crossing at the next adjacent interface $\lambda_i$. For each thermodynamic condition $T/P$, three independent FFS runs are carried out to obtain the geometric mean rate $\langle R \rangle = (\prod_{i=1}^{3} R_i)^{1/3}$ and the standard error of $\ln\langle R \rangle$ is given by $\sqrt{\sum_{i=1}^{3} (\ln R_i - \ln\langle R \rangle)^2 / 3}$[59].

In FFS, the milestones $\lambda_i$ are defined based on the size of the largest crystalline cluster in the melt. Three types of carbon atoms, namely, liquid-like, diamond-like, and graphite-like, are differentiated through a local bond-order parameter $q_l$[44]. Similar to the Steinhardt bond-order parameter[60], $q_l$ is rotationally invariant; however, as a local order parameter, $q_l$ reflects the inherently local character of nucleation. A diamond-like carbon is defined as the carbon atom that has a $q_6 > 0.5$ (see Fig. S8) and exactly 4 nearest neighbours, whereas a graphite-like carbon is the one that has a $q_3 < -0.85$ (see Fig. S8) and exactly three nearest neighbours. These choices are made based on both the distributions of $q_n$ for the three phases and the characteristic local coordination in diamond and graphite. For the $q_l$ analysis, the cutoff distance in identifying the nearest neighbours in both diamond and graphite is set to be 1.8 Å. In determining the size of a crystalline cluster through the connectivity analysis, the same cutoff distance 1.8 Å is used for diamond, whereas a greater cutoff distance 3.5 Å is used for graphite, to account for the inter-layer spacing between neighbouring graphene sheets.

### Interfacial tension calculations

Due to the thermal fluctuations, the interface between distinct phases is not entirely flat. According to capillary wave theory, these fluctuations can be connected to the interfacial tension in the following manner:

$$\gamma \equiv \frac{k_B T}{2\pi \langle \sigma^2 \rangle} \ln \frac{L}{\xi}, \tag{4}$$

where $T$ is the absolute temperature, $k_B$ is the Boltzmann constant, $\sigma^2$ is the mean squared fluctuation of molecules at the interface, $L$ is determined by the size along the $x$- or $y$- dimension (assuming $z$ is normal to the surface) and $\xi$ is the bulk correlation length[61]. Using this relationship, we are able to estimate the interfacial free energy differences between the liquid-graphite (basal facet) and the liquid-graphite (prismatic facet), using the phase coexistence MD simulations used to compute the liquid-solid phase boundary. We set $L = 38.046$ Å according to our simulation cell size, and $\xi = 1.9$ Å corresponding to the first minimum of the radial distribution function of liquid carbon. $\sigma^2$ is calculated as: $\langle \sigma^2 \rangle \equiv \langle (z - \langle z \rangle)^2 \rangle$ where we selected atoms from the largest liquid carbon cluster belonging to one of the two interfaces in the periodic cell. The calculation is halted when the chosen interface becomes locally vertical (i.e., the approximate tangent plane contains $\hat{z}$) or makes contact with the other interface. We carried out the comparison between the two facets for three simulations, each held at 10 GPa with varying temperatures (4600 K, 4700 K, and 4800 K).

## Data availability

The data that support the findings of this study are available open-source at https://doi.org/10.5281/zenodo.15567723. The implementation of forward flux sampling in LAMMPS is available at https://github.com/WanyuZhao/FFS-LAMMPS. Source data are provided with this paper.

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

## Acknowledgements

We are grateful to Gabriele C. Sosso for providing a critical assessment of the manuscript. This work was supported by the National Science Foundation under Grants No. 2053235 (DD) and 2053330 (TL).

## Author contributions

D.D. and T.L. conceived the study. D.D. and W.Z. conducted the molecular dynamics simulations. S.C. trained the machine learning models. W.Z., M.L.B., S.C., D.D., and T.L. analysed the results. D.D. drafted the manuscript. All authors contributed to editing and finalising the manuscript.

## Competing interests

The authors declare no competing interests.
