## [Transparent Peer Review file · Nature Communications]

Metastability and Ostwald Step Rule in the Crystallisation of Diamond and Graphite from Molten Carbon

Corresponding Author: Professor Davide Donadio

Version 0:

Reviewer comments:

Reviewer #1

(Remarks to the Author)

The article discusses the nucleation of diamond and graphite in supercooled liquid carbon under various pressure-temperature conditions using molecular dynamics with a machine-learning potential. One of the key highlights of the paper is the nucleation of metastable graphite at pressures where diamond is expected to form—an effect explained in terms of two-step nucleation. This result is, to say the least, quite fascinating.

However, from my perspective:

1) The authors do not provide sufficient evidence.

2) The result could easily be an artifact of the MD model rather than a genuine physical process.

I hope to be wrong in my second statement and invite the authors to address the following questions:

1) One of the main claims of the paper is that graphite nucleates beyond its region of thermodynamic stability. However, the authors do not calculate or define the boundaries of this region according to their ML potential. Instead, they limit their discussion to the rather enigmatic phrase in the caption of Fig. 1:

“The purple dashed line is our theoretical estimate of the graphite/diamond phase boundary.”

No explanation is provided regarding the basis of this “theoretical estimate”. If the actual graphite stability region for the given ML potential extends up to 14–15 GPa at the discussed temperatures, then the result is quite trivial, and there is no need to invoke two-step nucleation.

2) The authors trained their model on the dataset from the original GAP-2017, obtained via LDA-DFT [reference 43 in the bibliography]. However, LDA is known to poorly describe liquid and amorphous carbon. Gábor Csányi and his team have since transitioned to more advanced exchange functionals in their later models for a reason. For example, consider the difference between LDA and PBE predictions for the density of liquid carbon—see Fig. 2 in [<https://pubs.aip.org/aip/jap/article/116/1/013510/139206>]
[https://pubs.aip.org/aip/jap/article/116/1/013510/139206]—which reaches several dozen percent at certain pressures. Properly describing the liquid phase is crucial for the phase transition under discussion. Underestimating the density of liquid carbon could easily shift the graphite nucleation region to higher pressures.

3) The integration timestep—0.5 fs—is quite large for the temperatures discussed. Typical values in similar studies range from 0.1 to 0.2 fs:

0.1 fs in "Comparative study of melting of graphite and graphene"

0.2 fs in "Non-equilibrium melting and sublimation of graphene simulated with two interatomic potentials"

0.24 fs in "Carbon under extreme conditions: Phase boundaries and electronic properties from first-principles theory"

A large timestep can lead to poor energy conservation, which in turn could cause density fluctuations in liquid. Unfortunately, the authors do not provide any information on energy conservation at this timestep.

4) Many technical details are missing. The authors do not provide links to their model or dataset, neither in the manuscript nor on arXiv. Their description of the two-phase modeling is limited to a single paragraph, whereas such calculations typically warrant entire papers. Additionally, the aforementioned “theoretical estimate of the graphite/diamond phase

boundary” lacks elaboration. Since SI is unrestricted in length, I encourage the authors to provide all necessary data to reproduce their results. In its current state, the study is simply non-reproducible due to a lack of essential information.

5) The statement regarding the validation of the potential with experimental data:

“Our simulations successfully reproduce the experimental phase diagram and validate the accuracy of the machine learning potential utilized in this study.”

is nearly nonsensical, given the significant discrepancies among experimental results on the carbon phase diagram. E.g. you can find experimental studies predicting graphite melting temperatures in the range of 6300–6700 K

[<https://journals.aps.org/prl/abstract/10.1103/PhysRevLett.122.175702>] as well as in the range of 4000–4500 K (see the review in Carbon [<https://doi.org/10.1016/j.carbon.2004.12.027>]). In this context, nearly any potential—including REBO and Stillinger-Weber—could be considered in excellent agreement with the experimental phase diagram.

If the authors could reproduce the effect using a better-benchmarked ML potential (e.g., GAP-20) or perform a more in-depth analysis of their own potential, focusing on the carbon phase diagram parameters it predicts, I would be happy to withdraw my concerns. At present, however, the entire result could easily be an artifact of the MD potential, and the reader has no means to verify it. The history knows a number of cases when poorly validated carbon MD potentials tricked the researchers [<https://journals.aps.org/prb/abstract/10.1103/PhysRevB.48.3591>].

Reviewer #2

(Remarks to the Author)

The article “Metastability and Ostwald Step Rule in the Crystallisation of Diamond and Graphite from Molten Carbon” by Davide Donadio et al applies molecular dynamics simulations based on first-principles machine learning potentials to reveal the microscopic mechanisms of diamond and graphite nucleation from liquid carbon. The most important finding of this article is that at high pressures (10.5 ~ 15 GPa) in the thermodynamically stable region of diamond, liquid carbon preferentially forms metastable graphite rather than diamond through non-classical nucleation paths, following Ostwald's step rule. The article may be of interest to a wide range of specialists working in the field of earth and planetary science, and materials manufacturing, and deserves publication in nature communications. But I have two concerns.

1. Liquid carbon contains different local atomic structures around the GDL triple point, it is necessary to verify that whether the NEP3 potential is suitable to model the structure of liquid carbon. In the supplementary materials, the structural and elastic properties of graphite and diamond are listed, the structural properties of liquid carbon should also be listed.

2. The authors claim that spontaneous crystallisation of diamond and graphite was observed in direct MD simulations of liquid carbon at constant pressure, in which the temperature was ramped from 5,000 K to 3,500 K in 25 ns, corresponding to a cooling rate of 60 K/ns. In Figure 1a, there are eight spontaneous crystallization points, and the authors should clearly show how the initial liquid was obtained and whether the liquid reached equilibrium in the eight isobaric simulations, especially the three points between 10.0 and 15.0 GPa. This could clarify whether the findings were related to the initial liquid structure.

3. In Page 8, Figure 2(a-c) should be Figure 2(d-f).

4. Reference 23 is incomplete.

Reviewer #3

(Remarks to the Author)

The authors performed a range of simulations to study the crystallisation of carbon, particularly around the graphite-diamond-liquid triple point, where experimental measurements are inconclusive. To enable accurate behaviour of the model, they trained a machine learning based potential, and studied the phase behaviour and transitions, as well as the formation of critical nuclei under different pressure conditions. The manuscript is clearly written, the calculations appear to be carefully conducted with appropriate conditions drawn. I thus recommend the manuscript to be published, with a few minor details to be potentially considered by the authors:

The authors use the spherical harmonic expansion of the atomic environment as order parameter to identify crystalline environments in the simulation - are these the same as the Steinhardt bond order parameters widely used by the community? (<https://doi.org/10.1103/PhysRevB.28.784>)

The authors mention that reproducing the negative gradient of the graphite melting line is a challenge and most empirical models, or even ML potentials struggle to reproduce that. While I agree, there is some evidence that the highly accurate EDIP potential [<https://doi.org/10.1103/PhysRevB.63.035401>] might be able to capture this challenging feature (as shown in Ref 24 of the manuscript). Would the authors be able to comment on that?

In case when graphite structures were formed during the nucleation or quenching process, have the authors considered checking or evaluating the different relative orientation or stacking of the graphite sheets (i.e. how the hexagonal rings are positioned relative to each other on different layers)? While the ground state structure is relatively well described, at higher temperatures a range of various orientations or stackings should appear, each with slightly different free energy - would it be possible to determine if certain arrangements are more likely to be formed or more likely to facilitate the graphite-diamond transition?

The authors tested different radii for the various descriptors of their ML model, but I couldn't find the actual values they used. For graphite, where van de Waals interactions are important, and the atomic distance between layers is larger than within layers - this could be an important factor (as the authors also partially point out in their description). Could do authors present more details on this aspect of their model (maybe in the SI)?

In case of two-phase coexistence simulations, what was the orientation of the graphite structure compared to the liquid interface? One would expect that especially the growth of the solid phase is rather different if the graphite sheets are parallel to the interface than perpendicular. Have the authors considered this?

Version 1:

Reviewer comments:

Reviewer #1

(Remarks to the Author)

The authors have conducted substantial additional computational work to address the questionable aspects of the original draft and have published a comprehensive SI. In my opinion, the article is now suitable for publication in its current form.

P.S.

Please do not encourage the carbon community to blindly train ML models on low-accuracy DFT functionals such as LDA. While you have performed a thorough comparison with OptB88, many others may not be as cautious. As you can see from the phase diagram, the pressures along the graphite–diamond coexistence line have shifted by a factor of $\times 2$ - $\times 3$. Thank you for your understanding :)

(Remarks on code availability)

Reviewer #2

(Remarks to the Author)

The data supplemented by the authors dispelled my doubts. It can be seen from the supplementary data that during the spontaneous crystallization simulation, the atomic coordination numbers in all the initial liquids are mainly in triple coordination, and the structure of the liquid changes continuously with the pressure. I think it can be published. However, for the integrity of the data, I think that in Figs. S3-S5 the author should supplement another data that is very close to 15 GPa in Fig. 1 and Fig. S2. Although their initial states were very close, after isobaric cooling, one spontaneously crystallized into a diamond structure and the other spontaneously crystallized into a graphite structure. I think this is very important.

(Remarks on code availability)

Reviewer #3

(Remarks to the Author)

The Authors have considered the range of comments and feedback from reviewers. They performed more benchmarking calculations, provided more details on their analysis, and have also done some further free energy calculations to further validate their conclusions.

I recommend the paper to be accepted for publication.

(Remarks on code availability)

I have looked at the github page. I haven't tried to install the packages and run the example input files, but there seem to be comprehensive step-by-step instructions and input files for the LAMMPS simulations, all arranged in an easy to navigate structure.

RESPONSE TO REVIEWERS' COMMENTS

Reviewer #1 (Remarks to the Author):

The article discusses the nucleation of diamond and graphite in supercooled liquid carbon under various pressure-temperature conditions using molecular dynamics with a machine-learning potential. One of the key highlights of the paper is the nucleation of metastable graphite at pressures where diamond is expected to form—an effect explained in terms of two-step nucleation. This result is, to say the least, quite fascinating.

However, from my perspective:

1) The authors do not provide sufficient evidence.
2) The result could easily be an artifact of the MD model rather than a genuine physical process. I hope to be wrong in my second statement and invite the authors to address the following questions:

1) One of the main claims of the paper is that graphite nucleates beyond its region of thermodynamic stability. However, the authors do not calculate or define the boundaries of this region according to their ML potential. Instead, they limit their discussion to the rather enigmatic phrase in the caption of Fig. 1:

“The purple dashed line is our theoretical estimate of the graphite/diamond phase boundary.”

No explanation is provided regarding the basis of this “theoretical estimate”. If the actual graphite stability region for the given ML potential extends up to 14–15 GPa at the discussed temperatures, then the result is quite trivial, and there is no need to invoke two-step nucleation.

We have calculated the graphite/diamond coexistence line using Gibbs-Duhem integration (Kofke 1993, Ref. 30) starting from the triple point estimated as the crossing between the graphite/liquid and the diamond/liquid melting lines (pressure = 10.3 GPa, temperature = 4640 K).

This procedure is explained in the revised manuscript. Figure 1 has been updated, including the calculated points (maroon triangles) along the graphite/diamond coexistence line.

2) The authors trained their model on the dataset from the original GAP-2017, obtained via LDA-DFT [reference 43 in the bibliography]. However, LDA is known to poorly describe liquid and amorphous carbon. Gábor Csányi and his team have since transitioned to more advanced exchange functionals in their later models for a reason. For example, consider the difference between LDA and PBE predictions for the density of liquid carbon—see Fig. 2 in [\[https://pubs.aip.org/aip/jap/article/116/1/013510/139206\]](https://pubs.aip.org/aip/jap/article/116/1/013510/139206)—which reaches several dozen percent at certain pressures. Properly describing the liquid phase is crucial for the phase transition under discussion. Underestimating the density of liquid carbon could easily shift the graphite nucleation region to higher pressures.

We have performed simulations using neuroevolution potential fitted to the optB88-vdW dataset in Rowe et al. (Ref. 28) (NEP@OptB88-vdW). We were reluctant to use this potential in the first place because in the phase diagram published in Marchant et al. *npj Computational Materials* (2023)9:131 the slope of the graphite/liquid coexistence line near the triple point is positive, whereas there is convincing experimental evidence that it should be negative [see, for example, Ref. 5].

In the revised manuscript, we have characterized the liquid state with both the NEP@LDA and NEP@OptB88-vdW potentials and repeated the calculation of the phase diagram of carbon in the pressure range from 5 to 30 GPa with this model. The two models give very similar results except for a pressure offset of about ~6 GPa, as it can be seen in this figure where the solid lines represent the NEP@LDA phase diagram shifted to ~6 GPa higher pressure.

We have also repeated the crystallization studies (quenching from 5000 K to 3500 K) with NEP@OptB88-vdW, reproducing the metastable crystallization of graphite above the graphite-diamond coexistence line.

These results are now commented in the manuscript, and the comparison between the two phase diagrams, including the temperatures of spontaneous crystallization, is reported in Figure S2.

3) The integration timestep—0.5 fs—is quite large for the temperatures discussed. Typical values in similar studies range from 0.1 to 0.2 fs:

0.1 fs in "Comparative study of melting of graphite and graphene"

0.2 fs in "Non-equilibrium melting and sublimation of graphene simulated with two interatomic potentials"

0.24 fs in "Carbon under extreme conditions: Phase boundaries and electronic properties from first-principles theory"

A large timestep can lead to poor energy conservation, which in turn could cause density fluctuations in liquid. Unfortunately, the authors do not provide any information on energy conservation at this timestep.

This is a very valid point. We checked energy conservation in microcanonical (NVE) MD simulations of a 4096-atom model of liquid carbon. For example, at T=4500 K and P=7 GPa,

and the total energy fluctuation is of the order of 1 meV/atom without any significant energy drift over 100 ps. This leads to no significant drift in temperature. The total energy per atom of liquid carbon is shown below for a 100-ps NVT equilibration run followed by a 100-ps NVE run. The insert zooms in on the total energy of the microcanonical run.

To further address the reviewer's concern, we have repeated 4 quenching simulations (12, 14.5, 15.5, and 20 GPa) at the same quenching rates as the previous simulations with a timestep of 0.25 fs. The crystallization outcomes are the same, and the temperatures of spontaneous nucleation are within the statistical uncertainty range.

Energy conservation is discussed in the methods section of the revised manuscript.

4) Many technical details are missing. The authors do not provide links to their model or dataset, neither in the manuscript nor on arXiv. Their description of the two-phase modeling is limited to a single paragraph, whereas such calculations typically warrant entire papers. Additionally, the aforementioned "theoretical estimate of the graphite/diamond phase boundary" lacks elaboration. Since SI is unrestricted in length, I encourage the authors to provide all necessary data to reproduce their results. In its current state, the study is simply non-reproducible due to a lack of essential information.

We have created an archive with the simulation setups, the parameters of the NEP potentials, the version of the GPUMD program used for these simulations, and the Python script to run Gibbs-Duhem integration, and forward flux sampling code together with FFS input and example runs. The data are publicly available at <https://github.com/ddonadio/CarbonNucleation> and <https://github.com/WanyuZhao/FFS-LAMMPS>, and will be deposited in MaterialsCloud if the paper is accepted.

5) The statement regarding the validation of the potential with experimental data:

“Our simulations successfully reproduce the experimental phase diagram and validate the accuracy of the machine learning potential utilized in this study.”

is nearly nonsensical, given the significant discrepancies among experimental results on the carbon phase diagram. E.g. you can find experimental studies predicting graphite melting temperatures in the range of 6300–6700 K

[<https://journals.aps.org/prl/abstract/10.1103/PhysRevLett.122.175702>] as well as in the range of 4000–4500 K (see the review in Carbon [<https://doi.org/10.1016/j.carbon.2004.12.027>]). In this context, nearly any potential—including REBO and Stillinger-Weber—could be considered in excellent agreement with the experimental phase diagram.

We thank the reviewer for bringing to our attention more experimental material on the phase diagram of carbon. While we agree that there are large experimental uncertainties in the determination of the phase diagram of carbon, we may still argue that our model is more general and physically grounded than REBO and Stillinger-Weber that do not even predict graphite stability or suggest exotic liquid-liquid phase transitions in liquid carbon.

The results by Kondratyev&Rakhe PRL 2019 push the melting point of graphite at much higher temperatures than all the previous experiments. Similarly, Liang et al. [Phys. Rev. Research 1, 033090, (2019)] suggest that the melting temperature of diamond at 15 GPa should be higher (~6000 K) than previous estimates. These works are now cited to put our results in a better context (Refs. 7 and 8).

If the authors could reproduce the effect using a better-benchmarked ML potential (e.g., GAP-20) or perform a more in-depth analysis of their own potential, focusing on the carbon phase diagram parameters it predicts, I would be happy to withdraw my concerns. At present, however, the entire result could easily be an artifact of the MD potential, and the reader has no means to verify it. History knows a number of cases when poorly validated carbon MD potentials tricked the researchers [<https://journals.aps.org/prb/abstract/10.1103/PhysRevB.48.3591>].

The paper mentioned by the reviewer dates back to 1993 and uses a macroscopic model. It is not surprising that its conclusions about a liquid-liquid phase transition, although plausible, turned out to be incorrect. The same conclusions were reached by Glosli and Ree using the Brenner potential (REBO) (Ref. 22)

Following the reviewer’s suggestion, we have recalculated the phase diagram and repeated the spontaneous crystallization runs with a NEP potential fitted on the same dataset as the GAP-20 (based on OptB88-vdW data). Although we notice a shift of the graphite/diamond coexistence line to higher pressure, these new calculations reproduce the same features as the original study, including the metastable crystallization of graphite in the region of stability of diamond. The phase diagram thus obtained, including the temperatures at which we observe spontaneous crystallization of graphite and diamond, is shown in Figure S2 of the SI. The results are discussed in the main text, in the section on the phase diagram and spontaneous crystallization.

In passing, we need to point out that our results differ from the phase diagram published by Marchant et al. [npj Comput. Mater. Sci. 2023], which were obtained with simulations of very small cells, a method subject to much larger statistical uncertainties.

Reviewer #2 (Remarks to the Author):

The article “Metastability and Ostwald Step Rule in the Crystallisation of Diamond and Graphite from Molten Carbon” by Davide Donadio et al applies molecular dynamics simulations based on first-principles machine learning potentials to reveal the microscopic mechanisms of diamond and graphite nucleation from liquid carbon. The most important finding of this article is that at high pressures (10.5 ~ 15 GPa) in the thermodynamically stable region of diamond, liquid carbon preferentially forms metastable graphite rather than diamond through non-classical nucleation paths, following Ostwald's step rule. The article may be of interest to a wide range of specialists working in the field of earth and planetary science and materials manufacturing, and deserves publication in Nature Communications. But I have two concerns.

1. Liquid carbon contains different local atomic structures around the GDL triple point, it is necessary to verify that whether the NEP3 potential is suitable to model the structure of liquid carbon. In the supplementary materials, the structural and elastic properties of graphite and diamond are listed, the structural properties of liquid carbon should also be listed.

A detailed analysis of the structure of liquid carbon has been added to the paper (Figures S3, S4, and S5 in the Supporting Information). These properties are consistent with previous studies. In particular, the NEP models reflect the differences previously observed in the structure and density of liquid carbon between LDA and GGA functionals (see references 32, 33 and 36).

These differences, which lead to quantitative differences in the phase diagram, may be reconciled by shifting either the pressure or the density. Except for this pressure offset, the physical behavior of the liquid is qualitatively the same in the two models. A direct comparison with experiments suggests that the correct equation of state of the liquid is between those calculated (see Figure S3).

2. The authors claim that spontaneous crystallisation of diamond and graphite was observed in direct MD simulations of liquid carbon at constant pressure, in which the temperature was ramped from 5,000 K to 3,500 K in 25 ns, corresponding to a cooling rate of 60 K/ns. In Figure 1a, there are eight spontaneous crystallization points, and the authors should clearly show how the initial liquid was obtained and whether the liquid reached equilibrium in the eight isobaric simulations, especially the three points between 10.0 and 15.0 GPa. This could clarify whether the findings were related to the initial liquid structure.

The quenching simulations are initiated from liquid models equilibrated for at least 2 ns at 5000 K at each pressure. These are the same models that have been used to characterize the structure of liquid carbon.

3. In Page 8, Figure 2(a-c) should be Figure 2(d-f).

We have corrected the reference to the figure.

4. Reference 23 is incomplete.

We checked all the references.

Reviewer #3 (Remarks to the Author):

The authors performed a range of simulations to study the crystallisation of carbon, particularly around the graphite-diamond-liquid triple point, where experimental measurements are inconclusive. To enable accurate behaviour of the model, they trained a machine learning based potential, and studied the phase behaviour and transitions, as well as the formation of critical nuclei under different pressure conditions. The manuscript is clearly written, the calculations appear to be carefully conducted with appropriate conditions drawn. I thus recommend the manuscript to be published, with a few minor details to be potentially considered by the authors:

The authors use the spherical harmonic expansion of the atomic environment as order parameter to identify crystalline environments in the simulation - are these the same as the Steinhardt bond order parameters widely used by the community?

(<https://doi.org/10.1103/PhysRevB.28.784>)

These order parameters are derived from the original formulation in the Steinhardt paper; however, the version used here is "local," i.e., it provides a per-atom order parameter, whereas in the original Physical Review B article, Q_6 was defined as a global quantity. The necessity for local order parameters in nucleation studies was recognized in several works around the early 2000s. The exact formulation of the order parameter in this paper is given in Li et al. JCP 2009. To help the readers, we have added a brief description in the Methods section.

The authors mention that reproducing the negative gradient of the graphite melting line is a challenge and most empirical models, or even ML potentials struggle to reproduce that. While I agree, there is some evidence that the highly accurate EDIP potential (<https://doi.org/10.1103/PhysRevB.63.035401>) might be able to capture this challenging feature (as shown in Ref 24 of the manuscript). Would the authors be able to comment on that?

On page 4, we have added a note on the phase diagram computed with EDIP. We also noticed that, while the original LCBOP polarizable model gives a positive slope, the revised version LCBOPII gives a more reasonable melting curve of graphite. We have added a comment and a reference about this, too.

In case when graphite structures were formed during the nucleation or quenching process, have the authors considered checking or evaluating the different relative orientation or stacking of the graphite sheets (i.e. how the hexagonal rings are positioned relative to each other on different

layers)? While the ground state structure is relatively well described, at higher temperatures a range of various orientations or stackings should appear, each with slightly different free energy - would it be possible to determine if certain arrangements are more likely to be formed or more likely to facilitate the graphite-diamond transition?

The nuclei form in a highly dynamic state at high temperature. Under these conditions, the free energy differences among different stacking arrangements are below the thermal energy. We have examined the structure of graphite near critical size obtained in our FFS sampling, from which we found graphite nuclei appear to adopt random stacking. This is not surprising as the stacking fault energy of graphite was determined to be 0.85 mJ/m^2 (Rob H. Telling & Malcolm I. Heggie (2003), *Philosophical Magazine Letters*, 83:7, 411-421), which is about 2 orders of magnitude smaller than thermal energy (3800K) for one 1 nm^2 area.

The authors tested different radii for the various descriptors of their ML model, but I couldn't find the actual values they used. For graphite, where van de Waals interactions are important, and the atomic distance between layers is larger than within layers - this could be an important factor (as the authors also partially point out in their description). Could do authors present more details on this aspect of their model (maybe in the SI)?

The NEP models used in this work are now provided in the data archive. We have tested potentials with 2-body radii from 4.2 to 8 Å. We did not observe any significant differences either in the phase diagram or in the crystallization behavior.

In case of two-phase coexistence simulations, what was the orientation of the graphite structure compared to the liquid interface? One would expect that especially the growth of the solid phase is rather different if the graphite sheets are parallel to the interface than perpendicular. Have the authors considered this?

We tested configurations with the liquid in contact with the basal plane (0001) and the prismatic armchair plane (11-20) of graphite. We have chosen to use the prismatic plane because it has a much faster growth and melting kinetics, thus reducing the cost of the simulations. We added a comment about this in the methods section and a figure (Fig. S7) showing both graphite-liquid and graphite-diamond two-phase configurations.

RESPONSE TO REVIEWERS' COMMENTS

Reviewer #2 (Remarks to the Author):

The data supplemented by the authors dispelled my doubts. It can be seen from the supplementary data that during the spontaneous crystallization simulation, the atomic coordination numbers in all the initial liquids are mainly in triple coordination, and the structure of the liquid changes continuously with the pressure. I think it can be published. However, for the integrity of the data, I think that in Figs. S3-S5 the author should supplement another data that is very close to 15 GPa in Fig. 1 and Fig. S2. Although their initial states were very close, after isobaric cooling, one spontaneously crystallized into a diamond structure and the other spontaneously crystallized into a graphite structure. I think this is very important.

We agree with the reviewer that showing there is no discontinuity in the structural properties of the liquid at the tipping pressure of diamond nucleation is important. We have added data to Figures S3 and S5 at $P=15.5$ GPa for the NEP@LDA model and $P=19$ GPa for the NEP@OptB88-vdW model, which show that there is indeed no discontinuity either in the density or in the coordination of the liquid. We are reluctant to add the 15.5 GPa and the 19 GPa lines to the RDF figures (S4) because they are nearly indistinguishable from the 15 and 20 GPa lines, respectively.

Figures S3 and S5 and their captions have been updated with this information.